



# Augmented Kalman filter with a reduced mechanical model to estimate tower loads on an onshore wind turbine: a digital twin concept

Emmanuel Branlard[1], Dylan Giardina[1], and Cameron S. D. Brown[2]

[1]National Renewable Energy Laboratory, Golden, CO, USA
[2]Ørsted, Nesa Allé 1, 2820 Gentofte, Denmark

**Correspondence:** E. Branlard (emmanuel.branlard@nrel.gov)

**Abstract.** The paper presents an application of the Kalman filtering technique to estimate loads on a wind turbine. The approach combines a mechanical model and a set of measurements to estimate signals that are not available in the measurements, such as: wind speed, thrust, tower position, and tower loads. The model is several fold faster than real-time and is intended to be run online, for instance, to evaluate real-time fatigue life consumption of a field turbine using a digital twin. The mechanical model is built using a Rayleigh-Ritz approach and a set of joint coordinates. The paper presents a general method and illustrates it using a 2 degrees of freedom model of a wind turbine, and, using rotor speed, generator torque, pitch, and tower-top acceleration as measurement signals. The different components of the model are tested individually. The overall method is evaluated by computing the errors in estimated tower bottom equivalent moment from a set of simulations. From this preliminary study, it appears that the tower bottom equivalent moment is obtained with about 10% accuracy. The limitation of the model and the required steps forwards are discussed.

**DOF** degrees of freedom

**KF** Kalman filter

**FA** Fore-Aft

**TT** tower-top

**TB** Tower bottom

**SS** Side-side

## 1 Introduction

Wind turbines are designed and optimized for a given site using numerical tools, and, a statistical assessments of the environmental conditions the turbine will experience. The uncertainty on the tools and data are accounted for using multiplicative safety factors, which are determined from a combination of experience and specifications by the standards. Overconservative



safety factors will imply unnecessary costs which may be later on alleviated by extended the life time of a project. An under estimate of the safety factor will likely lead to catastrophic failures. Once a design is complete and the product is in place, is it possible to predict what the life time of the wind turbine will be?

Digital twins are becoming increasingly popular to follow the life cycle of a physical system. This concept is used to bridge the gap between the modelling and measurement realm: real time measurement from the physical system are communicated to a digital system and this information is combined with a numerical model to estimate the state of the system and potentially predict its evolution. A Kalman filter (KF) is an example of technique that can be used: it combines a model of a system with a set of measurements on this system to predict additional variables, such as positions or loads at points where no measurements are available. Other approaches are for instance: inverse methods, or neural network methods.

Kalman filters have been extensively used in control engineering with a wide range of applications. Auger et al. (2013) provide a review of some industrial applications. In the context of wind energy, wind speed estimation is critical for the determination of the dynamics of the system. This topic was for instance investigated using parametric models by Bozkurt et al. (2014), using Kalman filters by Østergaard et al. (2007), Knudsen et al. (2011), or Song et al. (2017), and using Luenberger type observer by Hafidi and Chauvin (2012). A comparison of wind speed estimation technique is found in Soltani

et al. (2013). The techniques were extended to also estimate the wind shear and turbine misalignments (see e.g. Bottasso et al. (2010), Simley and Pao (2016), Bertelè et al. (2018)). Kalman filtering has been used to estimate rotor loads and wind speed in application to rotor controls by Boukhezzar and Siguerdidjane (2011). Kalman filtering was recently used by Belloli (2019) to estimate the sea state based on the knowledge of the offshore platform position. More general approaches use Kalman filtering in combination with a model of the full wind turbine dynamics. These approach were used for wind speed estimation and

load alleviation via individual pitch control (Selvam et al. (2009), Bottasso and Croce (2009)), and for online estimation of mechanical loads Bossanyi (2003). An example of estimating tower loads with the acceleration sensor is for instance found in the report of M. (2008). Bossanyi et al. compared the observed rotor and tower loads with measurements, and investigated the potential of the control method to reduce damage equivalent loads (Bossanyi et al., 2012)

The methodology presented in this article uses an augmented KF (Lourens et al. (2012)) to estimate loads on the wind turbine

based on measurement signals commonly available in the nacelle. The method builds on the approach used by Bossanyi et al. and Lourens et al. The method of Lourens et al. is generalized. On the other hand, the expression of the mechanical system may be seen as simplified compared to the approach of Bossanyi et al.: a Rayleigh-Ritz formulation is used and the system is not further linearized. The equations are given in full for a 2 degrees of freedom system, and the source code is made available online. The time series of estimated loads are applied to assess the fatigue life consumption of the turbine components. The

study focuses on the determination of tower loads of onshore wind turbine. The numerical model of the wind turbine relies on a Rayleigh-Ritz shape-function approach with reduced numbers of degrees of freedom (Branlard (2019)). The wind speed is estimated using an approach similar to Østergaard et al. (2007), and the thrust force estimation is based on this wind speed estimate. The generator torque, the rotor speed, and the tower-top accelerations are used as measurements and combined with the numerical model with an augmented KF. The time series of loads in the tower are determined based on the tower shape

function and the tower degrees of freedom, and the fatigue loads are computed from this signal.





The first part presents the different components required for this work: the augmented KF, the numerical model of the turbine, and the estimators for the wind speed, thrust, tower load, and fatigue. Simple illustrations and validation results for the different components of the model are provided in a second part. The third part presents full applications but limited to simulations. Discussions and conclusions follow.

## 2   Description of the models

### 2.1   Example for a 2DOF wind turbine model

We start this section by an illustrative example, before describing the different parts of the model in their general form. A wind turbine is here modelled using 2 DOF: 1) the generalized coordinate associated with the fore-aft (FA) bending of the tower, $q_t$; 2) the shaft rotation, $\psi$. The tower bending is associated with a shape function $\Phi_t(z)$, such that the FA displacement of a point, at height $z$, and at time $t$, is given by $u(z,t) = q_t(t)\Phi_t(z)$. The shape function is normalized to unity at the tower-top, and $q_t$ is then equal to the FA displacement at the tower-top (see Figure 1). The equations of motion of the system are:

$$\begin{bmatrix} M & 0 \\ 0 & J \end{bmatrix} \begin{bmatrix} \ddot{q}_t \\ \ddot{\psi} \end{bmatrix} + \begin{bmatrix} C & 0 \\ 0 & 0 \end{bmatrix} \begin{bmatrix} \dot{q}_t \\ \dot{\psi} \end{bmatrix} + \begin{bmatrix} K & 0 \\ 0 & 0 \end{bmatrix} \begin{bmatrix} q_t \\ \psi \end{bmatrix} = \begin{bmatrix} T_a^* \\ Q_a - Q_g \end{bmatrix} \tag{1}$$

where: $M$, $C$, $K$ are the generalized mass, damping and stiffness associated with the FA DOF; $J$ is the drivetrain inertia; $T_a^*$ and $Q_a$ are the aerodynamic thrust and torque; and $Q_g$ is the generator torque. A star is used as upper-script of the thrust to indicate that using the thrust directly is a rough approximation. A more elaborate expression of the generalized force acting on $q_t$ is given in subsection 3.3. The determination of $M$, $C$ and $K$ is discussed in Branlard (2019). For the NREL-5MW turbine, the values are: $M = 4.4e^5$ kg, $D = 2.5e^4$ kg/s, $K = 2.7e^6$ kg/s² and $J = 4.3e^7$ kg.m². In this example, the system of equations is only coupled via the aerodynamic loads.

The following measurements are usually readily available on any commercial wind turbine: the generator power, $P_g$; the blade pitch angle, $\theta_p$; the rotor rotational speed, $\Omega \triangleq \dot{\psi}$. and the tower-top acceleration in the FA direction, $\ddot{q}_t$; The knowledge of the generator power, speed and losses allows to estimate the generator torque $Q_g$. In this study, the generator torque is assumed known. We will use an augmented KF concept to combine these measurements with the mechanical model to estimate the state of the system. The KF algorithm requires linear state and output equations. The state vector is assumed to be $x = [q_t, \psi, \dot{q}_t, \dot{\psi}, Q_a]$. The fact that some of the loads were included into the state vector is referred to as state augmentation. The choice of loads to include in the state vector is not unique and will lead to different state equations. Using this choice for $x$, Equation 1 is written into the following state equation:

$$\begin{bmatrix} \dot{q}_t \\ \dot{\psi} \\ \ddot{q}_t \\ \ddot{\psi} \\ \dot{Q}_a \end{bmatrix} = \begin{bmatrix} 0 & 0 & 1 & 0 & 0 \\ 0 & 0 & 0 & 1 & 0 \\ -M^{-1}K & 0 & -M^{-1}C & 0 & 0 \\ 0 & 0 & 0 & 0 & J^{-1} \\ 0 & 0 & 0 & 0 & 0 \end{bmatrix} \begin{bmatrix} q_t \\ \psi \\ \dot{q}_t \\ \dot{\psi} \\ Q_a \end{bmatrix} + \begin{bmatrix} 0 & 0 & 0 \\ 0 & 0 & 0 \\ M^{-1} & 0 & 0 \\ 0 & -J^{-1} & 0 \\ 0 & 0 & 0 \end{bmatrix} \begin{bmatrix} T_a^*(\dot{\psi}, Q_a, \theta_p) \\ Q_g \\ \theta_p \end{bmatrix} \tag{2}$$



where for simplicity the time derivatives of the aerodynamic torque is assumed to be zero, an assumption referred to as "random walk" force model. This assumption accounts of saying that the estimate of the torque at the next time step is likely to be close

to the one at the current time step. Improvements on this will be discussed in section 5. The thrust is determined based on the rotor speed, the aerodynamic torque, and the pitch angle, using tabulated data, as described in subsection 2.4. The output equation relates the measurements to the states and inputs as follows:

$$
\begin{bmatrix} \ddot{q}_t \\ \dot{\psi} \\ Q_g \\ \theta_p \end{bmatrix} = \begin{bmatrix} -M^{-1}K & 0 & -M^{-1}C & 0 & 0 \\ 0 & 0 & 0 & 1 & 0 \\ 0 & 0 & 0 & 0 & 0 \\ 0 & 0 & 0 & 0 & 0 \end{bmatrix} \boldsymbol{x} + \begin{bmatrix} M^{-1} & 0 & 0 \\ 0 & 0 & 0 \\ 0 & 1 & 0 \\ 0 & 0 & 1 \end{bmatrix} \begin{bmatrix} T_a^* \\ Q_g \\ \theta_p \end{bmatrix}
\tag{3}
$$

Equation 2 and Equation 3 are used within a KF algorithm to estimate the states vector based on the measurements. The

estimated time series of $q_t$, together with its associated shape function $\Phi_t$, are used to determine the bending moments within the tower and estimate tower fatigue loads, based on the method presented in subsection 2.5. Results from this simple model will be provided in section 3. The remaining paragraphs of this section generalize the approach presented.

## 2.2   Mechanical model of the wind turbine

The wind turbine is described using a set of degrees of freedom (DOF) that consist of joints coordinates and shape functions

coordinates. The method was described in previous work (Branlard, 2019), and the source code made available online, via a library called YAMS (Branlard). Similar approaches are for instance used in the elastic codes Flex and OpenFAST (OpenFAST). The advantage of the method is that the system can be described with few DOF. The number of DOF is between 2 and 30 DOF whereas traditional finite element methods require in the order of one thousand DOF.

    The only joint coordinate retained in the current model is the shaft azimuthal position, noted $\psi$. The shaft torsion, and nacelle

yaw and tilt joints can be added without difficulty. The tower and blades are represented using a set of shape functions taken as the first mode shapes of these components. The shape functions of the tower are assumed to be the same in the FA and side side (SS) directions, which are respectively aligned with the $x$ and $y$ directions (see Figure 1). The number of shape functions are noted $n_{xt}$, $n_{yt}$ and $n_b$ for the tower FA, tower SS, and blade respectively. Writing $B$ the number of blades, $n_s$ the number of DOF representing the shaft, the total number of DOF is: $n_q = n_s + Bn_b + n_{xt} + n_{yt}$. The tower DOF are written $q_{xt,i}$ with

$i \in [1..n_{xt}]$ and $q_{yt,i}$ with $i \in [1..n_{yt}]$. Similar notations are used for the blade DOF.

    The equation of motions are established using Lagrange's equation. The example presented in subsection 2.1 corresponds to $n_s = 1$, $n_b = 0$, $n_{t,SS} = 0$ and $n_{t,FA} = 1$. An example for a 5DOF system, with $n_s = 1$, $B = 2$, $n_b = 1$, $n_{t,SS} = 0$ and $n_{t,FA} = 1$, is given in Branlard (2019). In the general case, the equation of motions are described as:

$$
M\ddot{\mathsf{q}} + C\dot{\mathsf{q}} + K\mathsf{q} = \boldsymbol{f} \qquad \rightarrow \qquad \begin{bmatrix} \dot{\mathsf{q}} \\ \ddot{\mathsf{q}} \end{bmatrix} = \begin{bmatrix} \boldsymbol{0} & \boldsymbol{I} \\ -\boldsymbol{M}^{-1}\boldsymbol{K} & -\boldsymbol{M}^{-1}\boldsymbol{C} \end{bmatrix} \begin{bmatrix} \mathsf{q} \\ \dot{\mathsf{q}} \end{bmatrix} + \begin{bmatrix} \boldsymbol{0} \\ \boldsymbol{M}^{-1}\boldsymbol{f} \end{bmatrix}
\tag{4}
$$

where $\boldsymbol{M}$, $\boldsymbol{C}$, $\boldsymbol{K}$ are the mass, damping, and stiffness matrices; $\boldsymbol{q}$ is the vector of DOF; and $\boldsymbol{f}$ is the vector of generalized loads acting on the DOF. An inconvenience of the method is that the mass matrix is a non-linear function of the DOF. The





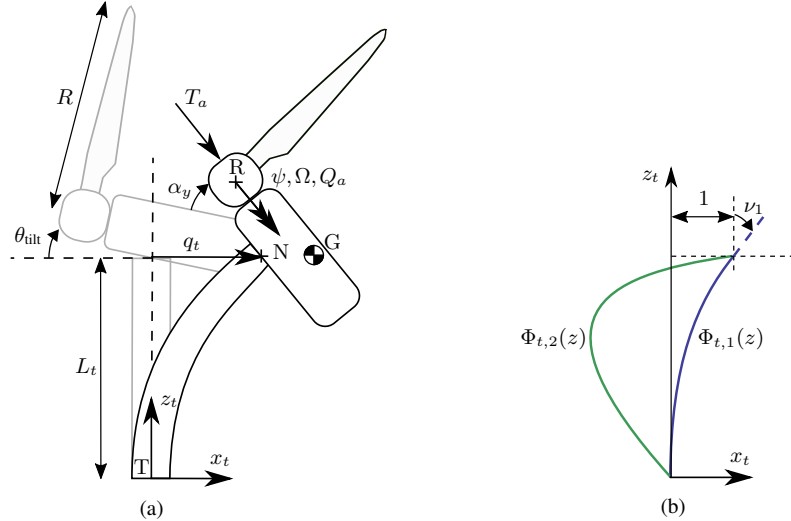

(a)                                                    (b)

**Figure 1.** (a) Notations for the wind turbine model and (b) example of shape functions used for the tower. Definition of points: T, tower bottom; N, tower-top; G, center of mass of the rotor nacelle assembly; R, rotor center. The shape functions are normalized to unity at point $N$. The slope at the extremity of the first shape function is written $\nu_1 \triangleq \frac{\mathrm{d}\Phi_{t,1}}{\mathrm{d}z}(L_t)$.

main assumption of this work is that the non-linearities can be discarded as a first approximation. This assumption is further discussed in section 5.

### 2.3 Augmented Kalman filter applied to a mechanical system

A description of the standard KF can be found e.g. in the textbook of Grewal and Andrews (2014) or Zarchan and Musoff (2015). The algorithm will not be detailed in this paper. The method expect a state and output equations of the following form:

$$\dot{\boldsymbol{x}} = \mathbf{X}_x \boldsymbol{x} + \mathbf{X}_u \boldsymbol{u} + \boldsymbol{w}_x \qquad \text{(state equation)} \qquad (5)$$

$$\boldsymbol{y} = \mathbf{Y}_x \boldsymbol{x} + \mathbf{Y}_u \boldsymbol{u} + \boldsymbol{w}_y \qquad \text{(output/measurement equation)} \qquad (6)$$

where $\boldsymbol{x}$, $\boldsymbol{u}$ and $\boldsymbol{y}$ are the state, input, and measurement vectors ; $\mathbf{X}_x$, $\mathbf{X}_y$, $\mathbf{Y}_x$ and $\mathbf{Y}_u$ are Jacobian matrices describing

the expected relationships between measurements, states and inputs ; and $\boldsymbol{w}_x$ and $\boldsymbol{w}_y$ are Gaussian uncorrelated noises associated with the state-space model and measurements respectively, of which the associated covariance matrices are noted $\boldsymbol{Q} = E[\boldsymbol{w}_x \boldsymbol{w}_x^t]$ and $\boldsymbol{R} = E[\boldsymbol{w}_y \boldsymbol{w}_y^t]$, with $E[\boldsymbol{w}_x \boldsymbol{w}_y^t] = 0$ and $E$ the expected value operator. We will develop these equations in the case of a mechanical system that follows the general form of Equation 4. Specific applications will be given in section 3 and section 4.

Different approaches can be used to write Equation 4 in the form of Equation 5, depending how the force vector is to be treated. In a first approach, the forces can be considered to be inputs $\boldsymbol{f} = \boldsymbol{u}$, in which case Equation 4 is directly in the form of Equation 5, with $\boldsymbol{x} = [\mathbf{q}, \dot{\mathbf{q}}]$. This implies that we have full knowledge of the forces acting on the system at every time step,





which is unlikely. In a second approach, the forces can be assumed to be part of the system noise, $\boldsymbol{w}_x$, which would lead to $\boldsymbol{x} = [\mathbf{q}, \dot{\mathbf{q}}]$, and $\boldsymbol{B} = 0$. This is obviously a crude approximation since the forces acting on the system are non stochastic, and,

we likely have some knowledge on them. In the intermediate approach introduced by Lourens et al. (Lourens et al., 2012), some of the forces are included in the system noise, and others as part of the states. The reduced set of loads that are part of the states is written $\boldsymbol{p}$, of length $n_p$, and the full force vector is assumed to be approximated by: $\boldsymbol{f} \approx \boldsymbol{S}_p \boldsymbol{p}$, where $\boldsymbol{S}_p$ is a matrix of dimension $n_q \times n_p$. The reduced set of forces, $\boldsymbol{p}$, is integrated into the state vector as: $\boldsymbol{x} = [\mathbf{q}, \dot{\mathbf{q}}, \boldsymbol{p}]$. This process is referred to as state augmentation.

We introduce a generalized approach and assume that the forces are a combination of states, inputs and unknown noise:

$$\boldsymbol{f} \approx \mathbf{F}_q \mathbf{q} + \mathbf{F}_{\dot{q}} \dot{\mathbf{q}} + \mathbf{F}_p \boldsymbol{p} + \mathbf{F}_u \boldsymbol{u} + \boldsymbol{w}_f \approx +\mathbf{F}_q \mathbf{q} + \mathbf{F}_{\dot{q}} \dot{\mathbf{q}} + \mathbf{F}_p \boldsymbol{p} + \mathbf{F}_u \boldsymbol{u} \tag{7}$$

where the $\mathbf{F}_{\bullet}$ matrix represent the Jacobian of the force vector with respect to vector $\bullet$, and $\boldsymbol{w}_f$ are unknown forces that are assumed to be part of system disturbance $\boldsymbol{w}_x$. The terms $\mathbf{F}_q$ and $\mathbf{F}_{\dot{q}}$ are linearized stiffness and damping terms. These terms are zero if their contributions are already included in the definitions of $\boldsymbol{K}$ and $\boldsymbol{C}$. In practice, the linearization of the force

vector may not be possible, and assumed relationships or engineering models are used. As an example, if $\boldsymbol{p}$ contain the thrust force and $\boldsymbol{f}$ the moment at the tower base, the appropriate element of $\mathbf{F}_p$ could be set with the lever arm between tower-top and tower base.

This approach allows us to use the knowledge we have of some of the main loads acting on the system and express their dynamics into the state-space equation. The forces may for instance be assumed to follow a first order system as follows:

$$\dot{\boldsymbol{p}} = \mathbf{P}_q \mathbf{q} + \mathbf{P}_{\dot{q}} \dot{\mathbf{q}} + \mathbf{P}_p \boldsymbol{p} + \mathbf{P}_u \boldsymbol{u} \tag{8}$$

where the $\mathbf{P}_{\bullet}$ matrices are obtained from a knowledge of the force evolution. Second order system could also be introduced, in which case the state needs to be augmented with both $\boldsymbol{p}$ and $\dot{\boldsymbol{p}}$ ("random walk" force model). For simplicity, the applications used in this work will assume $\dot{\boldsymbol{p}} = \boldsymbol{0}$, but future work will investigate the benefit of using first order systems for the evolution of the forces.

Inserting Equation 7 into Equation 4, introducing $\boldsymbol{x} = [\mathbf{q}, \dot{\mathbf{q}}, \boldsymbol{p}]$, and using Equation 8, a state equation of the form of Equation 5 is obtained:

$$\mathbf{X}_x = \begin{bmatrix} \mathbf{0} & \boldsymbol{I} & \mathbf{0} \\ -\boldsymbol{M}^{-1}(\boldsymbol{K} - \mathbf{F}_q) & -\boldsymbol{M}^{-1}(\boldsymbol{C} - \mathbf{F}_{\dot{q}}) & \boldsymbol{M}^{-1}\mathbf{F}_p \\ \mathbf{P}_{\dot{q}} & \mathbf{P}_q & \mathbf{P}_p \end{bmatrix}, \qquad \mathbf{X}_u = \begin{bmatrix} \mathbf{0} \\ \boldsymbol{M}^{-1}\mathbf{F}_u \\ \boldsymbol{M}^{-1}\mathbf{P}_u \end{bmatrix} \tag{9}$$

The measurements are assumed to be a combination of the acceleration, velocity, displacements, loads and inputs:

$$\boldsymbol{y} \approx \mathbf{Y}_{\ddot{q}} \ddot{\mathbf{q}} + \mathbf{Y}_{\dot{q}} \dot{\mathbf{q}} + \mathbf{Y}_q \mathbf{q} + \mathbf{Y}_p \boldsymbol{p} + \tilde{\mathbf{Y}}_u \boldsymbol{u} \tag{10}$$

Inserting the acceleration $\ddot{\mathbf{q}}$ from Equation 4 into Equation 10, an output equation of the form of Equation 6 is obtained, with:

$$\mathbf{Y}_x = \begin{bmatrix} \mathbf{Y}_q - \mathbf{Y}_{\ddot{q}} \boldsymbol{M}^{-1}\boldsymbol{K}, & \mathbf{Y}_{\dot{q}} - \mathbf{Y}_{\ddot{q}} \boldsymbol{M}^{-1}\boldsymbol{C}, & \mathbf{Y}_p + \mathbf{Y}_{\ddot{q}} \boldsymbol{M}^{-1}\mathbf{F}_p \end{bmatrix}, \qquad \mathbf{Y}_u = \tilde{\mathbf{Y}}_u + \mathbf{Y}_{\ddot{q}} \boldsymbol{M}^{-1}\mathbf{F}_u \tag{11}$$





Equation 9 and Equation 11 form the bridge between the definition of the mechanical model and the state and output equations needed by the KF algorithm.

Equation 5 and Equation 6 are in continuous form, whereas the KF algorithm uses discrete forms. The discrete form of the matrices perform the time integration of the states from one time step to the next, namely: $\boldsymbol{x}_{k+1} = \mathbf{X}_{x_d}\boldsymbol{x}_k + \mathbf{X}_{u,d}\boldsymbol{u}_k$, where the subscript $d$ indicates the discrete form of the matrices and $k$ is the time step index. The matrix $\mathbf{X}_{x,d}$ is referred to as the fundamental matrix. For time-invariant systems, this matrix may be obtained using Laplace tranform or by Taylor-series expansion (Zarchan and Musoff, 2015). For a given time step $\Delta t$, the discrete matrices corresponding to $\mathbf{X}_x$ and $\mathbf{X}_u$ are:

$$\boldsymbol{X}_{x,d} = e^{\boldsymbol{X}_x\Delta t} = I + \boldsymbol{X}_x\Delta t + \frac{(\boldsymbol{X}_x\Delta t)^2}{2!} + \ldots \approx I + \boldsymbol{X}_x\Delta t \tag{12}$$

$$\boldsymbol{X}_{u,d} = \int_0^{dt} \mathbf{X}_{x,d}(\tau)\boldsymbol{X}_u\,\mathrm{d}\tau \approx [\mathbf{X}_{x,d} - \boldsymbol{I}]\,\mathbf{X}_x^{-1}\mathbf{X}_u \approx \mathbf{X}_u\Delta t$$

The approximation in Equation 12 is effectively a first order forward Euler time integration. The matrix $\mathbf{Y}_x$ and $\mathbf{Y}_u$ remain unchanged by the discretization since the ouput equation is an algebraic equation involving quantities at the same time step.

Many choices are possible as to how the model may be formulated: which forces should be accounted for in the reduced set $\boldsymbol{p}$, which forces should be assumed to be obtained from the inputs, which models to use for the $\mathbf{P}$ matrices, etc. Since the study is limited to onshore wind turbines, the main loads are the aerodynamic thrust and torque. A subtlety to account for, is that some of the forces of the model presented in Equation 4 are generalized forces, which are projection of loads onto the shape functions (Branlard, 2019). An example will be given in subsection 3.3.

The Jacobian matrices introduced should be determined by linearization about an operating point. The mass matrix should also be linearized about such point. In the current work, the non-linearities are either neglected, or directly inserted into the expression presented without performing a linearization. This crude simplification will be discussed in section 5, in light of the results presented in section 3 and section 4.

### 2.4 Wind speed and thrust estimation

In this paragraph, $Q_a$, $\theta_p$ and $\Omega$ are assumed to be given. The aerodynamic power and thrust coefficients, $C_P$ and $C_T$, are also assumed to be known as function of the pitch angle and tip speed ratio, $\lambda = \Omega R/U_0$, where $R$ is the rotor radius and $U_0$ the wind speed. The functions $C_P(\lambda,\theta_p)$ and $C_T(\lambda,\theta_p)$ are estimated by running a parametric set of simulations at constant operating conditions. Some uncertainty is here present as to whether the real turbine does performs as predicted by these functions. This question will be considered in section 5. The aerodynamic torque is computed from the tabulated data as:

$$Q_{a,\text{tab}}(U_0,\Omega,\theta_p) = \frac{1}{2}\rho\pi R^2\frac{U_0^3}{\Omega}\,C_P\left(\frac{\Omega R}{U_0},\theta_p\right) \tag{13}$$

The wind speed is obtained by solving the following non-linear constraint equation for $u_{\text{est}}$:

Find $u_{\text{est}}$, such that $Q_a - Q_{a,\text{tab}}(u_{\text{est}},\Omega,\theta_p) = 0$ (14)





where $\rho$ is the air density, which is another potential source of uncertainty to be considered when dealing with measurements. The wind speed determined by this method is assumed to be the effective wind speed acting over the rotor area. A correction for nacelle displacements is discussed in section 5. The aerodynamic thrust is estimated from this wind speed as:

$$T_{a,\mathrm{est}} = T_{a,\mathrm{tab}}\left(u_{\mathrm{est}}, \Omega, \theta_p\right), \qquad \text{with} \quad T_{a,\mathrm{tab}}\left(U_0, \Omega, \theta_p\right) = \frac{1}{2}\rho\pi R^2 U_0^2\, C_T\left(\frac{\Omega R}{U_0}, \theta_p\right) \tag{15}$$

### 2.5 Tower loads and fatigue estimation

The deflection of the tower, $U$, in the $x$ or $y$ directions, at a given height $z$, and a given time $t$, is given by the sum of the tower shape functions scaled by the tower degrees of freedom:

$$U_x(z,t) = \sum_i q_{xt,i}(t)\Phi_{t,i}(z), \qquad U_y(z,t) = \sum_i q_{yt,i}(t)\Phi_{t,i}(z) \tag{16}$$

The curvature, $\kappa$, is obtained by differentiating the deflection twice, giving:

$$\kappa_x(z,t) = \sum_i q_{xt,i}(t)\frac{\mathrm{d}^2\Phi_{t,i}}{\mathrm{d}^2 z}(z), \qquad \kappa_y(z,t) = \sum_i q_{yt,i}(t)\frac{\mathrm{d}^2\Phi_{t,i}}{\mathrm{d}^2 z}(z) \tag{17}$$

The bending moments along the tower heights are then obtained from the curvatures using Euler beam theory:

$$M_y(z,t) = EI(z)\kappa_x(z,t), \qquad M_x(z,t) = EI(z)\kappa_y(z,t) \tag{18}$$

where $EI$ is the bending stiffness of a given tower cross section. The time series of bending moment are processed using a rain flow counting algorithm to estimate the equivalent loads and damage (WG3, 2005).

## 3 Simple applications and validations

### 3.1 Wind speed estimation

The wind speed estimation methodology presented in subsection 2.4 is illustrated and evaluated in this section. Tabulated $C_P$ and $C_T$ values were obtained for the NREL-5MW turbine (Jonkman et al., 2009) using the multi-physics simulation tool OpenFAST (OpenFAST). A turbulent simulation was devised such as to sweep through the main operating regions of the wind turbine within a 10-min period, namely: the startup region (region 0), the optimal $C_p$ tracking region (region 1), rotor-speed regulation (region 2), and power regulation (region 3). For the NREL-5MW turbine region 2 has a small span and it is here gathered with region 3. The turbine was simulated with all the DOFs turned on. The following variables were extracted from the simulation at 50Hz: $\overline{u}_{\mathrm{ref}}$, the average wind speed at the rotor plane; $Q_{a,\mathrm{ref}}$, the aerodynamic torque; $T_{a,\mathrm{ref}}$ the aerodynamic thrust; $\Omega_{\mathrm{ref}}$ the rotational speed; and $\theta_{p,\mathrm{ref}}$ the pitch angle. The wind speed, $u_{\mathrm{est}}$, was estimated using the method presented in subsection 2.4. The results are presented in Figure 2 and commented below.

The absolute error in wind speed is mostly within $\pm 0.5$m/s, both values being indicated by two dashed lines on the left figure. The error is greatest in region 0 where the generator torque is not yet applied. A separate wind speed method should be

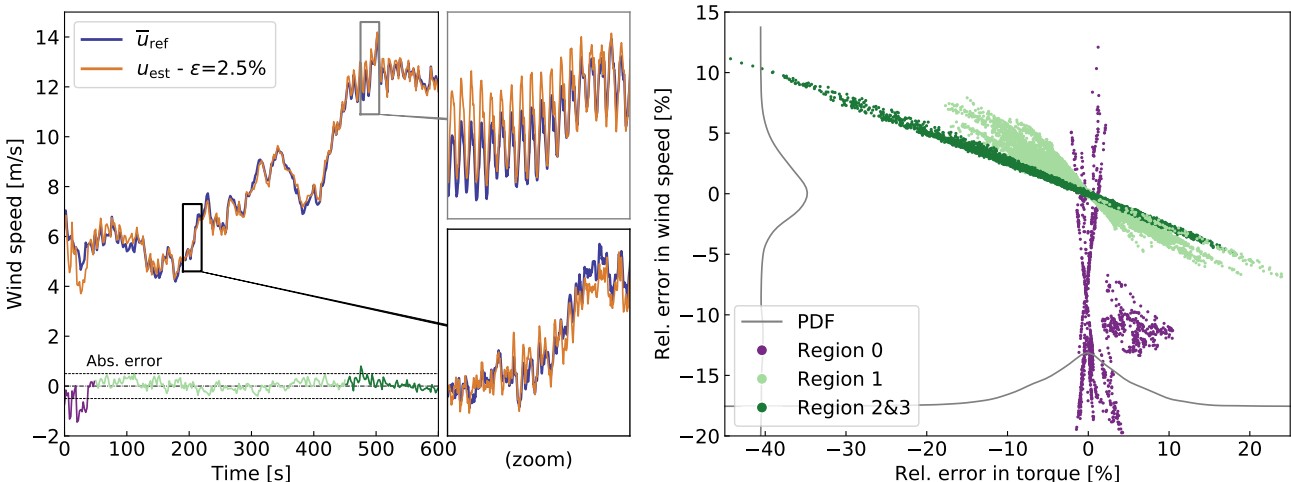

**Figure 2.** Wind speed estimation based on a reference simulation. (Left:) Wind speed time series and absolute error colored by operating regions. Two extracted windows are shown. (Right:) Relative error in estimated torque and wind speed colored by operating regions. The probability density function (PDF) is given for each axis.

devised for this case. The mean relative error for the entire time series is $\epsilon = 2.5\%$. The estimated wind speed is seen to follow the challenging trends of this time series, matching both the low and high frequencies. In the top zoom, it is seen that no phase

lag is observed in the estimated wind speed, but the estimated value is seen to overshoot.

There are several potential sources of errors in the current methodology. One concern is whether the unsteady aerodynamic torque, can be determined using a look-up table that uses instantaneous values. The relative error between the unsteady torque $Q_{a,\mathrm{ref}}$ and the tabulated torque, $Q_{a,\mathrm{tab}}(\overline{u}_{\mathrm{ref}}, \Omega_{\mathrm{ref}}, \theta_{p,\mathrm{ref}})$, is used as the $x$-axis on the right of Figure 2. A wide range of values is obtained, with the error varying between $-40\%$ and $20\%$. Such estimation of the torque is likely to be accurate only for

slow varying wind fields, where the effects of dynamic wake and dynamic stall on the blade loading will be limited. The tabulated method may be improved by accounting for these unsteady aerodynamic effects (discussed in section 5). Another question is whether the effective wind speed, that characterizes the aerodynamic forces on the turbine, is indeed the average wind speed at the rotor plane. The relative error between $\overline{u}_{\mathrm{ref}}$ and $u_{\mathrm{est}}$ is used as $y$-axis on the right figure. A large error on this axis may not necessarily indicate that the estimated wind speed is wrong, since indeed this estimated wind speed is such that

$Q_a = Q_{a,\mathrm{tab}}(u_{\mathrm{est}}, \Omega_{\mathrm{ref}}, \theta_{p,\mathrm{ref}})$, as a result of the minimization involved in Equation 14. The estimated wind speed may thus be expected to be different from the rotor averaged wind speed. During the startup period, the error in wind speed can be large and is uncorrelated to the error in torque, yet a stronger correlation is seen when the turbine is producing power. Looking at the probability density functions given in the right of Figure 2, it is seen that the errors in torque and wind speed are centered on zero. The fact that both errors are centered on zero, indicate that when the unsteady torque can indeed be obtained using

instantaneous values and tabulated data, the (estimated) effective wind speed is close to the average wind speed at the rotor. Other sources of errors are discussed in section 5.





A more thorough study on the questions raised above are left open for future work. Overall, the results from the test case are encouraging. Wind speed estimation is a standard feature of most wind turbine controllers, and it is likely that more advanced features are implemented by manufacturers. Any improvement on the methodology used in the current paper will be beneficial
for the procedure of loads estimation presented.

### 3.2   Thrust estimation

The wind speed estimated in subsection 3.1 is used to estimate the thrust $T_{a,\text{est}}$ with Equation 15. In Figure 3, the estimated thrust value is compared to the unsteady aerodynamic thrust from the simulation, $T_{a,\text{ref}}$. The values of $T_{a,\text{tab}}(\overline{u}_{\text{ref}}, \Omega_{\text{ref}}, \theta_{p,\text{ref}})$ are also shown in the figure. For this simulation, the thrust signal was obtained with a mean relative error of $1.5\%$ over the range of

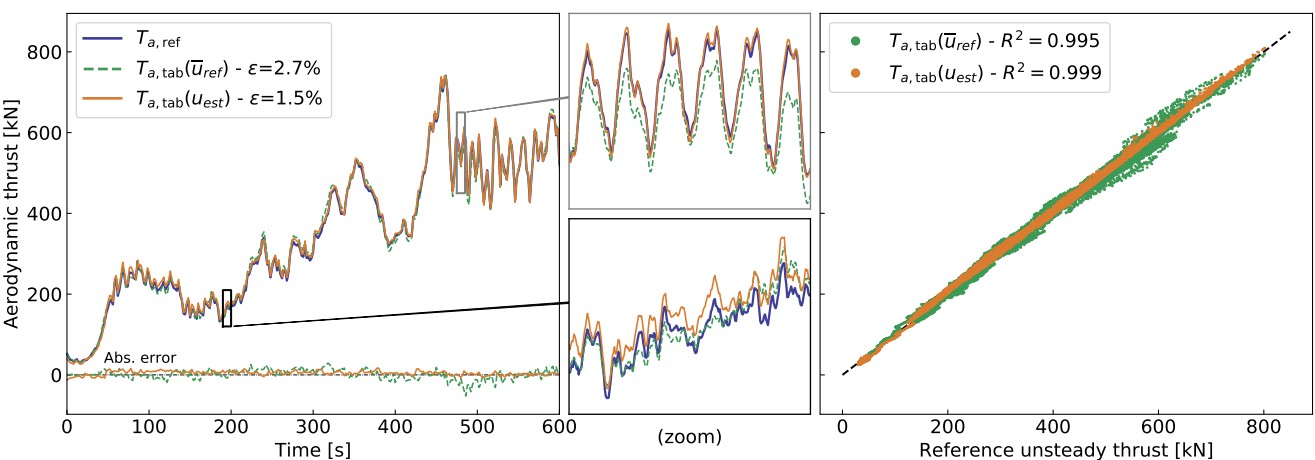

**Figure 3.** Comparison of aerodynamic thrusts: $T_{a,\text{ref}}$, obtained from a reference simulation; $T_{a,tab}(\overline{u}_{\text{ref}})$, obtained from tabulated $C_T$ and the rotor averaged wind speed from the simulation; $T_{a,\text{est}} = T_{a,\text{tab}}(u_{\text{est}})$, obtained from the estimated wind speed. (Left:) Time series of thrust and absolute errors compared to the $T_{a,\text{ref}}$. (Right:) Scatter plot of the tabulated thrust compared to the reference thrust.

operating conditions considered. Using the estimated wind speed is seen to produce thrust values closer to the reference thrust than if $\overline{u}_{\text{ref}}$ is used. In line with the discussions of subsection 3.1, this could support the fact that the estimated wind speed provides an effective velocity, consistent with the instantaneous state of the rotor, but different from the rotor averaged wind speed. Yet, it is also possible that compensating errors are at play, or, that the thrust is less sensitive to changes of wind speed or drive-train dynamics than the torque. Despite these open questions, we continue by assuming the method provide thrust
estimates with sufficient accuracy.

### 3.3   Reduced model of the mechanical system

The 2-DOF mechanical model presented in subsection 2.1 is here compared to the more advanced OpenFAST model consisting of 16 DOF. As mentioned in subsection 2.1, the generalized force acting on $q_t$ can be further improved. The notations from



Figure 1 are adopted. The resulting force and moment at the tower-top are written $\mathcal{F}_N$ and $\mathcal{M}_N$. The contribution of this

load to the generalized force is $\boldsymbol{f}_N = \boldsymbol{B}_N \cdot [\boldsymbol{F}_N; \mathcal{M}_N]$ where, according to the virtual work principle, $\boldsymbol{B}_N$ is the velocity transformation matrix that provides the velocity of point $N$ as function of other DOFs. More details on this formalism are provided in Branlard (2019). For the single tower DOF considered, the $B$-matrix consists of the end values of the shape function deflection and slope, i.e. $\boldsymbol{B}_N = [\Phi_{t,1}(L_t), 0, 0, 0, \nu_1, 0]$, where $L_t$ is the length of the tower and $\nu_1 \triangleq \frac{d\Phi_{t,1}}{dz}(L_t)$. The shape functions are assumed to be normalized at their extremity, i.e. $\Phi_{t,i}(L_t) = 1$, so that the generalized force is:

$$f_N = \mathcal{F}_{x,N} + \nu_1 \mathcal{M}_{y,N} \tag{19}$$

The main forces acting at the tower-top are assumed to the be aerodynamic thrust and the gravitational force from the rotor nacelle assembly (RNA) mass, $M_{\text{RNA}}$. The loads are then obtained as:

$$\mathcal{F}_{x,N} = T_a \cos(\alpha_y + \theta_{\text{tilt}}), \quad \mathcal{M}_{y,E} = T_a [x_{NR} \sin\theta_{\text{tilt}} + z_{NR} \cos\theta_{\text{tilt}}] + gM_{\text{RNA}} [x_{NG} \cos\alpha_y + z_{NG} \sin\alpha_y] \tag{20}$$

where: $\theta_{\text{tilt}}$ is the tilt angle of the nacelle; $NR$ is the vector from the tower-top to the rotor center, where the thrust is assumed

to act; $NG$ is the vector from the tower-top to the RNA center of mass; $g$ is the acceleration of gravity; and $\alpha_y$ is the $y$-rotation of the tower-top due to the bending of the tower (see Figure 1). For a single tower mode $\alpha_y(t) = q_t(t)\nu_1$. The linearization of Equation 19 and Equation 20 for small values of $q_t$ leads to:

$$f_N = q_t \left\{ -T_a \nu_1 \sin\theta_{\text{tilt}} + \nu_1^2 gM_{\text{RNA}} z_{NG} \right\} + (T_a \cos\theta_{\text{tilt}}) + T_a \nu_1 [x_{NR} \sin\theta_{\text{tilt}} + z_{NR} \cos\theta_{\text{tilt}}] + \nu_1 gM_{\text{RNA}} z_{NG} \tag{21}$$

where: the term in parenthesis is the main contribution, which justifies the use of $T_a$ in Equation 1; the term in curly brackets

is seen to act as a stiffness term. The presence of $T_a$ in this term introduce an undesired coupling and this term is kept on the right-hand-side of Equation 1. It is noted that the vertical force $\mathcal{F}_{z,N}$ contributes to the softening of the tower. The main softening effect attributed to the RNA mass is included in the stiffness matrix, as described in Branlard (2019). The contribution of the thrust to the softening, and additional contribution of quadratic velocity forces to the generalized force are neglected.

  The other elements of the 2D model are obtained from the OpenFAST input files. The mass, stiffness and damping matrix

of Equation 1 are obtained using the YAMS library (Branlard, 2019) which can take as input an OpenFAST model, and thus use the same shape functions. The damping of the 2 DOF model was tuned based on simple "decay" simulations, to include the aerodynamic damping contribution. The simulation used for validation consists of a linear ramp of wind speed from 0 to 10 m/s in the first 100 s, and a sudden drop to 6 m/s at 200s. The aerodynamic loads, and the generator torque are extracted from the OpenFAST simulation and applied as external forces to the reduced order model. Time series of tower-top positions,

rotational speed and tower bottom moments are compared in Figure 4. The rotational speed is well captured, indicating that the rotational inertia is properly set, but also indicating that the drive-train torsion does not have a strong impact for this simulation. The overall trend of the tower-top displacements is also well captured, though more differences are present due to missing contributions from additional blade and tower DOFs, missing non-linearities and quadratic velocity forces.

  The method from subsection 2.5 is used to estimate the bending moments along the tower from the tower-top displacement.

The results shown on the right of Figure 4 indicate that the overall trends and load levels are well estimated, but some offsets





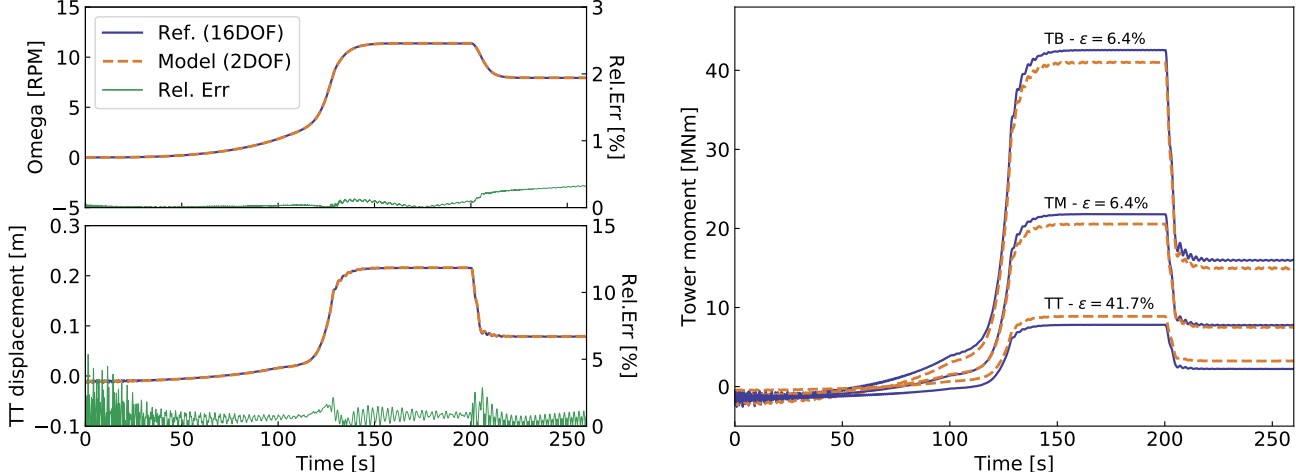

**Figure 4.** Simulation results using OpenFAST (16 DOF) and the reduced 2-DOF model. (Left): Rotational speed and tower-top (TT) displacements. (Right:) Tower moments at three different heights: tower bottom (TB), tower middle (TM), and tower-top (TT). The TB moment is taken at 5% height above the ground and not exactly at the ground.

are observed, which are function of height. A contribution to the moment may be missing in the current model. This will be taken into consideration when analysing the results from the KF analysis.

## 4   Application to wind turbine tower loads estimation

Some of the individual models presented in section 2 were briefly validated in section 3. The augmented KF described in
subsection 2.3 is now used, combining the different models together with the measurements. The state and output equations given in Equation 2 and Equation 3 are implemented. The state equation is discretized according to Equation 12 and provided to the KF algorithm. Results from the KF simulation, which combines a set of measurements with a model, will be referred to as "KF estimation". The values used for the covariance matrices, $P$ and $Q$, are discussed in section 5.

### 4.1   Ideal cases without noise

The same simulation as the one presented in subsection 3.1 is used, which extends from region 0 to region 3. The measurements sampled at 20 Hz are here directly taken from the OpenFAST simulation and not from a field experiment. This is obviously an ideal situation since no noise is present in the measurements. Further, the OpenFAST and KF models are based on the same parameters such as the mass and stiffness distribution. States and tower loads estimated using the KF model are compared with the simulation results in Figure 5. The signals are seen to be well estimated by the KF model over the entire range of operating
regions. The error observed for the tower bottom moment is in the range of errors observed for the isolated mechanical system (subsection 3.3).



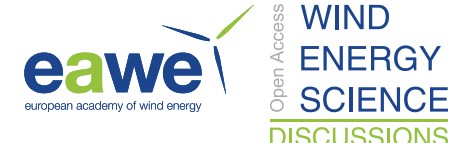

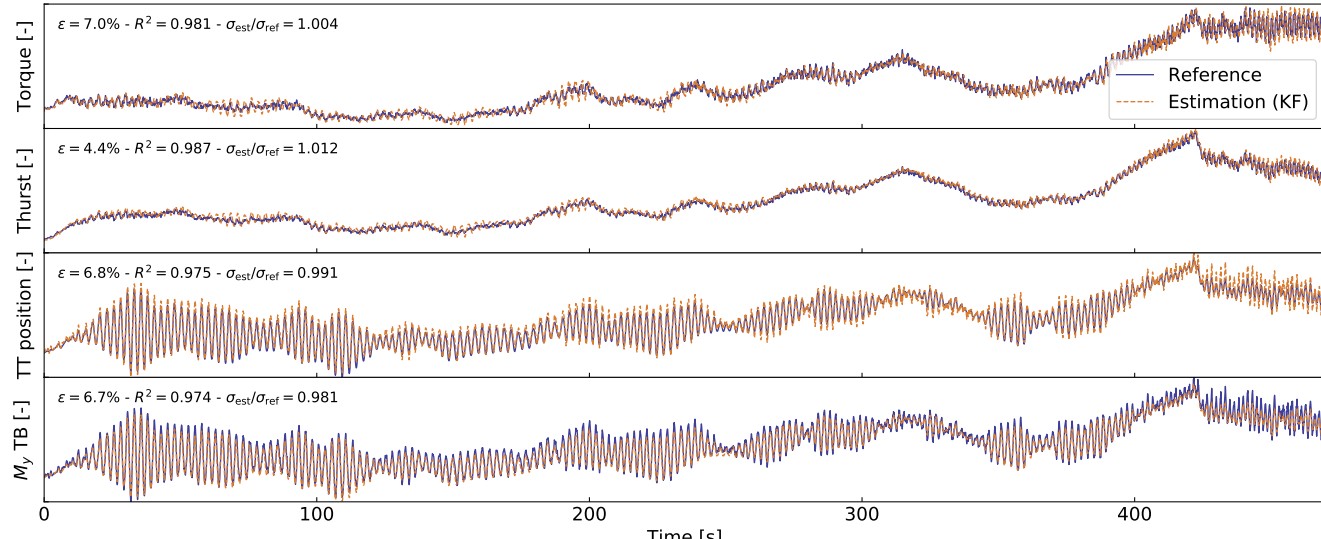

**Figure 5.** Comparison of signals simulated by OpenFAST (reference) compared with the ones estimated with the KF model. From top to bottom, dimensionless time series of: aerodynamic torque, aerodynamic thrust, tower-top displacement, Fore-Aft tower bottom moment.

A turbulent simulation is run, at an average wind speed of $14 \, \mathrm{m/s}$ with turbulence intensity of $0.14$, to illustrate the differences in the power spectral density of the signals. The results are given in Figure 6 and commented further. Frequencies that

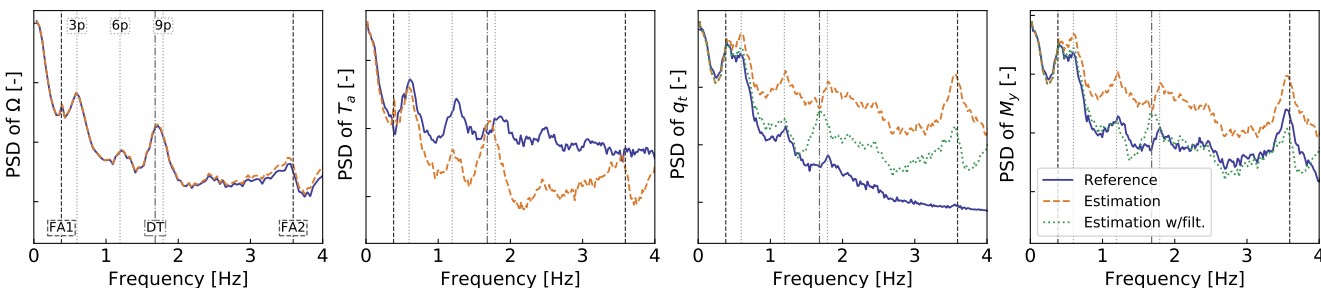

**Figure 6.** Power spectral density (PSD) of signals simulated by OpenFAST and estimated with the KF model for a turbulent simulation at 14 m/s. From left to right: rotational speed, Thrust, TT displacement and TB FA moment. Ticks on the y-axis represent two decades. The main system frequencies are marked with vertical lines: FA modes, drivetrain torsion (DT), and multiple of the rotational frequency $p = 0.2$.

are not in the mechanical system (e.g. the second FA mode and the DT torsion) are still "captured" by the estimator via the measurements. The rotational speed is directly observable by the KF, so the signal is obviously well estimated. The thrust, is estimated based on the rotational speed and thus exhibits similar frequencies as the rotational speed, which is not the case for the reference thrust signal. The integration of the acceleration into the TT position ($q_t$) shows a higher frequency content than the reference signal. The second FA frequency has a strong energy content in the estimated $q_t$ signal. This frequency content






comes from the acceleration signal, but it is not sufficiently captured and damped by the model which does not represent the
2nd mode. A moving average filter of period $1$ s was introduced to reduce the high-frequency content of the acceleration. The results are labelled "Estimation w/filt." on the figure. The analysis of the moment spectrum given on the right of Figure 6 indicates that the frequencies are well captured but the overall content at frequencies beyond the 1st FA mode is too high. This is indicated by the values of the equivalent loads which are respectively $20$ MNm and $30$ MNm for the reference and estimated signal, using a Wöhler slope of $m = 5$. The low-pass filter on the acceleration signal greatly improves the spectrum of $M_y$. The
error in equivalent loads is further quantifies in the next paragraph.

### 4.2 Simulations with noise

The simulations presented in subsection 4.1 used as measurements the simulated values from OpenFAST. In this section, a Gaussian noise is added to each of the OpenFAST signals in order to account for measurement uncertainties. The noise level is taken a 10% of the standard deviation of the signal simulated by OpenFAST. A noise level of $20\%$ will be referred to as "Large
noise". Simulations were performed with OpenFAST for 10 wind speeds, with six different turbulent seeds for each wind speed. A noise level was applied to these simulation results, prior to feeding them to the KF estimator. Cases with or without applying the low-pass filter on the (noisy) acceleration input were tried. Results for the error in equivalent load and standard deviation of the TB moment are shown in Figure 7. The equivalent loads are estimated using a Wöhler slope of $m = 5$. As expected,

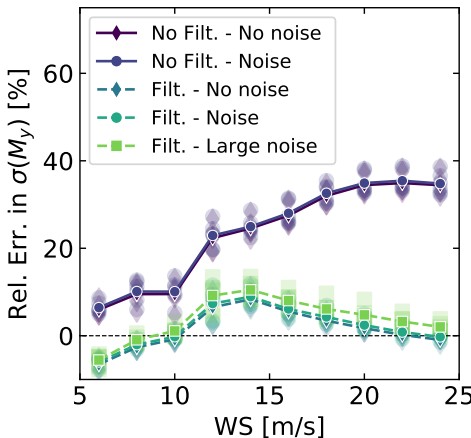
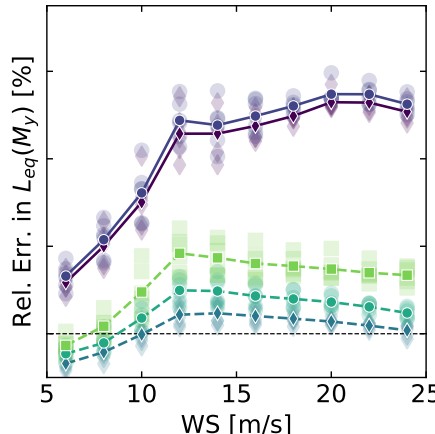

**Figure 7.** Comparison of the equivalent load and standard deviation of the TB moment as obtained by OpenFAST or as estimated from the KF estimator for different noise levels and with or without a filter on the acceleration input. (Left): Error in standard deviation. (Right:) Error in equivalent load. A positive value indicates that the estimator is overestimating. Individual markers indicate a simulation at a given wind speed and turbulence seed number. Lines indicate the mean values.

the errors in standard deviation and equivalent loads follow similar trends. Errors without filtering are several fold larger than
when the acceleration is filtered. Without noise, the equivalent loads are estimated with $\pm 8\%$ error. The error increases with



the noise level and the equivalent loads appear to be mostly overestimated. Further tuning of the filter and of the covariance matrices involved in the KF may reduce the error. Further discussions are provided in section 5.

## 4.3 Computational time

The framework is written in the noncompiled Python language. The code was run on a single CPU. The average computational
time for a 10 min period of measurements at 20 Hz was 37 s. Doubling the frequencies and the number of DOF would still keep the computational time several fold smaller than real time. The expensive part of the algorithm is the non-linear solve needed to find the optimal wind speed (Equation 14).

## 5 Discussions

**Measurements**  The results presented in the current study remained within the simulation realm. The accuracy of the method
under uncertain conditions was partly quantified using various noise models. Yet, future work will evaluate the model using field measurement data.

**Model choices**  As mentioned in subsection 2.3, a certain level of choice is present as to whether the loads are placed as an input or within the state vector. A consequence is that different load models may also be implemented, for instance, models of higher order that the one used in Equation 8. In the current study, a "random walk" force model was used for the torque, and
the thrust was set as a dependent variable of the torque. Yet, these loads are functions of the axial inductions, which typically are assumed to follow a second order model referred to as dynamic wake. A linearization of this model could be applied to the aerodynamic thrust and torque and potentially improve the performance prediction of the Kalman filter.

**Non-linearities and time-invariance**  This study assumed a linear form of the equation of motion and that the system matrices were time-invariant. Despite this crude assumption, reasonable results were obtained. Yet, further improvement are likely to be
obtained if these assumptions are lifted. A simple approach would consist in updating the system matrices at some given interval based on a slow moving average on the wind speed or the tower-top position. A more advanced method would use filtering methods that are adapted to non-linear systems, such as extended Kalman filters or particles filters. This approach would yet greatly increase the computational time. A shortcoming of the current approach is that the linear form of the equation was established "by hand". A more systematic approach will be considered in the future, using the linearized form of the state
matrices returned by OpenFAST, which would include aerodynamic damping directly.

**Degrees of freedom and offshore application**  The general formalism presented in section 2 can be applied to more degrees of freedom than the 2DOF model used: adding more shape function for the tower, and including side-side motion, yaw, tilt, shaft torsion, and blade motions. The results from the 2DOF model appeared encouraging enough to limit ourselves to this set, but future work will consider the inclusion of additional DOF. The extension of the method to offshore application could
be done by adding extra degrees of freedom for the substructure, or, by using shape functions that represent the entire support



structure. The generalized force due to the wave loading would need to be included. This force may be modelled based on the wind speed, or assumed of the model noise (see subsection 2.3).

**Model tuning**  Apart from the choices of degrees of freedom and model formulation, there remains a part of model tuning, through: the choice of covariance matrices, and, the potential filtering done on the measurements. As shown in subsection 4.2,

the filtering of the acceleration was seen to greatly improve the performance of the model. A time constant of $1s$ was chosen empirically for the filter, but this value may need to be adapted for other applications. The choice of values used for the covariance matrices is usually the main source of criticism for KF based models. Indeed, these values have a strong influence on the results, and they are usually tuned empirically. For the current method to be successfully applied on various wind plants, an automatic tuning procedure is required. In the current study, the covariance matrices of the process were set automatically

based on the value of the standard deviation of the simulated signal at rated conditions. For the measurements, these values were divided by two. It was found that this procedure lead to satisfactory results. A sensitivity study should be considered in future work to give further insight on the procedure, in particular if more states and measurements are used.

**Wind speed estimation and standstill/idling condition**  The wind speed estimation model presented in subsection 2.4 is limited to cases where the turbine is operating. Also, the accuracy of this model is crucial for the determination of the thrust,

which in turns determine the tower-top position and the tower loads. The nacelle velocity was for instance omitted in the current study and could be considered in future studies. Wind speed estimation is a field in which the industry has a great expertise. Improvements on the algorithm would benefit the model presented in this paper.

**Airfoil performance**  The performance of the airfoils is a large source of uncertainty which was not addressed. The thrust was determined using tabulated $C_T$ data, which may be significantly affected by the airfoil performance, which in turn are affected

by blade erosion or other roughness sources, and, additional uncertainty on the aerodynamic modelling. Further improvement of the model is thus required to provide an accurate determination of the thrust that would account for such unknowns. Air density should also be considered for a correct account of the loading if a tabulated approach is used.

## 6   Conclusions

The paper presented a general approach using Kalman filtering to estimate loads on a wind turbine, combining a mechanical

model and a set of readily available measurements. An open source framework was established in hope to be further applied for real-time fatigue estimation of wind turbine loads, providing inspiration for a digital twin concept. As an example, the equations for a 2DOF system of a wind turbine were presented, and this system was used throughout the article. The study focused on the estimation of tower bending moment and in particular the associated damage equivalent load. Based on simulation results, the estimator was seen to be able to capture the damage equivalent loads with an accuracy in the order of $10\%$. Future

work will address the following points: use of field measurements, offshore application of the method, increased number of DOF, automatic covariance tuning, improved wind speed estimation in standstill, improved thrust determination in off-design conditions, and use of a linearized model obtained from an aero-servo-elastic tool.



*Competing interests.*   No competing interest are present.





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
