# Peer review of "Augmented Kalman filter with a reduced mechanical model to estimate tower loads on an onshore wind turbine: a digital twin concept"

_Wind Energy Science, 2020_

## Referee Comment (RC1) · Martin Evans (Referee) · 30 Mar 2020

The subject is relevant and important, namely the estimation of turbine loads in real time. It is generally considered true that manufacturers are aiming to make their turbines smarter and this includes such technology. However, the abstract mentions real-time fatigue life consumption specifically, and it is this reviewer's opinion that no one needs to know fatigue life consumption in real-time. Perhaps other uses of real-time load estimation could help readers see the importance of the work.

The method is broadly applicable, to most turbine types onshore, with a discussion about the future offshore. The techniques used are explained well and are well under-

stood by engineers in this field. The validation method is good and honest - too often signal estimation is claimed to absurd accuracy. The conclusions accurately represent the body of the work.

The language used is appropriate, not being too heavy on mathematics that only mathematicians would know. Important areas of nuance and difficulty are expounded, rather than glossed over, which is refreshing. The paper could be shorter, but possibly at the expense of reproducibility by future readers. The references are relevant and cover the field reasonably well. It could be said that there are interesting load estimators that do not require KF, but perhaps the author wants to focus only on KF.

Proofreading notes: * Line 18 "Wind turbines are designed and optimized for a given site" - or are they designed to a class definition? * Line 42 has a citation to "M." - unless this is James Bond's boss, this might be a Latex error. * Line 49 talks about fatigue life consumption of "turbine components" - there is a huge variety in methods of FLC, so since this paper is only about rainflow counting for the tower, perhaps reword this to be less overreaching? * Line 55 makes it sound like thrust is all you need for the loads up the tower. Isn't rotor asymmetric loading from e.g. shear more significant as you go up the tower? * Line 160 inline equation $X_{x\_d}$ I think should be $X_{x,d}$. * Line 186 "where \rho" should be after equation (13) not (14) * Line 199 another citation error to WG3? * Line 355 previously you have amounts of time in the main text, not math mode so this time constant of 1s looks odd with the s italic.

---

## Referee Comment (RC2) · Anonymous Referee #2 · 2 May 2020

This paper presents: *An application of the Kalman filtering technique to estimate tower-base loads on a wind turbine using readily available measurements and a simple 2-DOF model, e.g., for use as a digital twin. *Validation of the method using OpenFAST simulations for a land-based wind turbine. *Recommendations for future work in this area.

Overall, the paper is well written, the results appear to be scientifically sound, and the results are informative. A few corrections and clarifications are warranted to approve the final publication. Please find specific comments and technical corrections below:

Specific Comments (Page / Line / Comment): 3 / 67 / Do the stiffness and damping

[Figure]

account for aerodynamic stiffness and damping, or just structural? I.e. do the aerodynamic stiffness and damping come from $T\_a^*$ or through C and K. 6 / 152 / The matrix "P"\_u" should not be premultiplied by $M^{-1}$. 6 / 154 / Why is Y\_qdd included? As indicated next, qdd can be derived from qd, q, p, and u, so, including Y\_qdd is redundant; the acceleration-related terms can be captured directly within Y\_qd, Y\_q, Y\_p, and Y\_u. 9 / 212 / Figure 2b is a bit hard to understand and could be clarified a bit more in the text. 11 / 258 / Is there a reason the inertia of the RNA is not accounted for in the model, i.e. a M\_RNA*qdd\_t term?

Technical Corrections (Page / Line / Comment): 2 / 21 / Change "extended" to "extending" 2 / 39 / Change "approach" to "approaches" 2 / 43 / Add a period at the end of the sentence. 5 / 119 / Change "X\_y" to "X\_u". 5 / 122 / Presumably "t" represents a transpose? Please clarify. 7 / 160 / Change "X\_x\_d" to "X\_x,d". 12 / 288 / Do you mean Q and R matrices, as used in section 2.3? 14 / 310 / Change "quantifies" to quantified".

---

## Author Comment (AC1) · 29 May 2020

Dear reviewers,

Thank you for your time and comments. We have replied to them in a separate document enclosed. We also enclose a "diff" between the current version of the manuscript and the previous one. We will attempt to shorten the manuscript and run a thorough proofreading before submitting a revised version.

Thanks again,

Emmanuel, Dylan and Cameron

[Figure]

Please also note the supplement to this comment:
https://www.wind-energ-sci-discuss.net/wes-2020-55/wes-2020-55-AC1-supplement.zip
* * *

---

## Author Response (AR1)

Dear reviewers,

Thank you so much for your feedback and for taking the time to review this article.

We have replied to your comments below. We will further work on trying to reduce the length of the paper and perform a thorough proofreading before submitting a revised version.

Again, we appreciate your time,

   Emmanuel, Dylan and Cameron

**Reviewer 1:**

The subject is relevant and important, namely the estimation of turbine loads in real time. It is generally considered true that manufacturers are aiming to make their tur-bines smarter and this includes such technology. However, the abstract mentions real-time fatigue life consumption specifically, and it is this reviewer's opinion that no one needs to know fatigue life consumption in real-time. Perhaps other uses of real-time load estimation could help readers see the importance of the work.
>>> Thank you for this useful comment. We have added that the method can also be applied to condition monitoring, or to develop dedicated control strategies.

The method is broadly applicable, to most turbine types onshore, with a discussion about the future offshore. The techniques used are explained well and are well understood by engineers in this field. The validation method is good and honest - too often signal estimation is claimed to absurd accuracy. The conclusions accurately represent the body of the work. The language used is appropriate, not being too heavy on mathematics that only mathematicians would know. Important areas of nuance and difficulty are expounded, rather than glossed over, which is refreshing. The paper could be shorter, but possibly at the expense of reproducibility by future readers. The references are relevant and cover the field reasonably well. It could be said that there are interesting load estimators that do not require KF, but perhaps the author wants to focus only on KF.
>>> We are indeed only focusing on KF methods, but to address your comments, we have now mentioned this limitation and added some references to other load estimation techniques (such as lookup table, machine learning, or modal expansion)

Proofreading notes:

* Line 18 "Wind turbines are designed and optimized for a given site" - or are they designed to a class definition?
>>> This is a fair point. We've added "class definition" in the sentence since site-specific (or position-specific) designs might not be the norm yet.

* Line 42 has a citation to "M." - unless this is James Bond's boss, this might be a Latex error.

>>> Thank you for noticing this and thank you for using humor!

* Line 49 talks about fatigue life consumption of "turbine components" - there is a huge variety in methods of FLC, so since this paper is only about rainflow counting for the tower, perhaps reword this to be less overreaching?

>>> The sentence now precise that rainflow counting is used, and the only component considered in this study is the tower.

* Line 55 makes it sound like thrust is all you need for the loads up the tower. Isn't rotor asymmetric loading from e.g. shear more significant as you go up the tower?

>>> Yes, it is true, the tilt and yaw loads will not be captured with only a tower top accelerometer. Yaw loads generate torsion on the tower so they are not so important for tower fatigue monitoring. But tilt loads add to the bending moment. Tilt moments are constant down the tower but thrust induced moments increase as you go down with the moment arm length. So the relative importance between tilt loading and thrust loading varies through the tower. At the tower bottom the tilt loads are pretty much insignificant contributor but at the tower top they are the main contributor. So the fatigue loads on the top part (e.g. 20-30%) of the tower would not be accurately estimated with the simple model presented in the paper. We added the following to the introduction "It is noted that the method is expected to be more accurate at the tower-bottom than the tower-top since rotor asymmetric loading is not captured."

* Line 160 inline equation $X_{x_d}$ I think should be $X_{x,d}$.
* Line 186 "where \rho" should be after equation (13) not (14)
* Line 199 another citation error to WG3?
* Line 355 previously you have amounts of time in the main text, not math mode so this time constant of 1s looks odd with the s italic.

>>> Thank you for noticing these errors, they have now been corrected.

**Reviewer 2:**

This paper presents:

*An application of the Kalman filtering technique to estimate tower-base loads on a wind turbine using readily available measurements and a simple 2-DOF model, e.g., for use as a digital twin.

*Validation of the method using OpenFAST simulations for a land-based wind turbine.

*Recommendations for future work in this area.

Overall, the paper is well written, the results appear to be scientifically sound, and the results are informative.  A few corrections and clarifications are warranted to approve the final publication. Please find specific comments and technical corrections below:

Specific Comments (Page / Line / Comment):

  3 / 67 / Do the stiffness and damping account for aerodynamic stiffness and damping, or just structural? I.e. do the aero- dynamic stiffness and damping come from T_a* or through C and K.
>> Thank you for this comment, we have now added some precision in the text. The damping C has been tuned to account for aerodynamic damping (using a decay test with the turbine operating). This damping is currently assumed to hold for all wind speeds, which is a rough approximation. The aerodynamic stiffness is part of T_a*, which is a quasti-steady aerodynamic load.

6 / 152 / The matrix "P"_u" should not be premultiplied by Mˆ-1."
>>> You are correct.

6 / 154 / Why is Y_qdd included?  As indicated next, qdd can be derived from qd, q, p, and u, so, including Y_qdd is redundant; the acceleration-related terms can be captured directly within Y_qd, Y_q, Y_p, and Y_u.
>>> You are correct, Y_qdd is introduced for convenience. If an accelerometer is present in the structure, and this measured acceleration is easily expressed in terms of the acceleration of the degrees of freedom, then the user can input this simple acceleration relationship into the matrix Y_qdd. Alternatively, the user can express the measured acceleration in terms of qd, q, p and u directly, in which case Y_qdd is not used. If an automated linearization procedure is used to determine the Y_* matrices, then Y_qdd should be skipped since it would be redundant. We now attempt to explain this in the text, using tilde notations for the output equation with acceleration.

9 / 212 / Figure 2b is a bit hard to understand and could be clarified a bit more in the text.
>>> Thank you for your comment. The figure was indeed hard to understand, and it was chosen to remove it altogether, since it was leading to a long discussion in the following paragraph. The main point was that the estimated wind speed can be different from the spatial average wind speed over the rotor, and this difference is not necessarily an "error". The estimated wind speed is, by the current definition, the best wind speed such that the tabulated aerodynamic torque matches the current unsteady aerodynamic torque. Another point was that region 0 needs a dedicated wind speed estimation, something which was already captured by the right figure. These are issues/remarks related to wind speed estimation and are digression for the topic of this article, so we chose to keep these discussions to

a minimum. We now only briefly mention these points. Hopefully, avoiding these discussions will clarify the paper and make it shorter.

11 / 258 / Is there a reason the inertia of the RNA is not accounted for in the model, i.e. a M_RNA*qdd_t term?
>>> The RNA inertia is part of the mass matrix term M given in equation 1. The following precision was added in section 3.3: "Velocity transformation matrices are used to convert individual component matrices (e.g. blades, nacelle) into the global system matrices. The mass matrix thereby comprise the inertia terms from the tower and RNA."

Technical Corrections (Page / Line / Comment):

2 / 21 / Change "extended" to "extend-ing"
2 / 39 / Change "approach" to "approaches"
2 / 43 / Add a period at the end of the sentence.
5 / 119 / Change "X_y" to "X_u".
5 / 122 / Presumably "t" represents a transpose? Please clarify.
7 / 160 / Change "X_x_d" to "X_x,d".
14 / 310 / Change "quantifies" to "quantified".
>>> Thank you for noticing these errors, they have been corrected now.

12 / 288 / Do you mean Q and R matrices, as used in section 2.3?
>>> You are correct, this was erroneous.

[revised manuscript text omitted]